# Fatty acid composition, total phenolic and total flavonoid contents, and antioxidant activity of Niger seed (*Guizotia abyssinica*) accessions collected from major producer areas of Ethiopia

**Megersa Chali Makuria[1,2], Amare Aregahegn Dubale[2,3], Minaleshewa Atlabachew[4], Marie Yayinie[5]***

1 Department of Chemistry, College of Natural and Computational Science, Dambi Dollo University, Dambi Dollo, Ethiopia, 2 Department of Chemistry, College of Natural and Computational Science, Dilla University, Dilla, Ethiopia, 3 Department of Chemistry, College of Natural and Computational Science, Addis Ababa University, Addis Ababa, Ethiopia, 4 Department of Chemistry, College of Science, Bahir Dar University, Bahir Dar, Ethiopia, 5 Department of Chemistry, College of Natural and Computational Science, Debre Tabor University, Debre Tabor, Ethiopia

* marieyayinie@gmail.com

**Data Availability Statement:** All relevant data are within the manuscript.

## Abstract

### Background

Oils from various sources are vital nutritional components with a variety of roles in our body. Niger seed (*Guzoita abyssinica*) is endemic to Ethiopia and is among the major oil seed crops grown in the country. The fatty acid composition and the concentration of other bioactive phytochemicals in it vary with species type, geographical origin, cultivation season, and varietal types. The present work investigated the fatty acid profile and the total phenolic (TPC), total flavonoid content (TFC) and antioxidant activity (AA) of Niger seed samples obtained from five different zones in the Amhara and Oromia regions of Ethiopia. using internationally accepted standard methods.

### Results

In all the samples, its main unsaturated acids were linoleic acid, ranging between 67.30 and 74.67% with respect to the relative percentage comprising 179 to 234 mg/g in terms of concentration; oleic acid constitutes between 5.43 and 11.02% of the total fatty acid or 1.03 and 1.60 mg/g of dry matter. Among saturated acids, it was the most abundant palmitic acid, ranging between 10.32 and 10.66% of the entire fatty acids comprising 24.80 to 37.10 mg/g. Amongst the zones, the seed from Amhara region, specifically from North Gondar has been the richest regarding a total of 347.74 mg/g. In addition, the total phenolic content ranged between 10.89 and 11.78 mg GAE/g, whereas the content of total flavonoids ranged from 5.42 to 6.67 mg CE/g. Aqueous methanol (80%) extracted more phenolic content than absolute methanol. On a regional basis, the Amhara region, represented by the North Gondar

**Funding:** The author(s) received no specific funding for this work.

**Competing interests:** The authors have declared that no competing interests exist.

and East Gojjam zones of the study area, had relatively higher TPC and TFC than other regions. The DPPH scavenging assay $IC_{50}$ value (μg/mL) ranged between 133–188 μg/mL and poorly correlated with TPC.

## Conclusion

Among the different fatty acids obtained, four of them, linoleic, oleic, palmitic, and stearic acids, are the major ones, followed by a significant amount of phenolic compounds irrespective of the variety of the studied samples and sampling of locations. The study also confirmed that TPC and TFC are not the only phytochemicals responsible for the antioxidant activity of the niger sees, as was reflected by the poor correlation between TPC and AA activity. Hence these findings indicate that the seeds of Niger could be an important source of essential fatty acids and medicinally important phytochemicals important for nutritional health improvement and agricultural development in Ethiopia.

## 1. Introduction

Edible oils are vital nutritional components with multiple roles in our body, such as sources of energy, membrane structures, body temperature regulation, and insulating organs [1, 2]. They are commonly used sources of various physiologically active fatty acids, carotenoids, phospholipids, natural antioxidants, and the fat-soluble vitamins E and K, depending on the species to which the oil belongs. It also has nonfood industrial benefits, such as cosmetics, paints, detergents, lubricants, and oleochemicals [3–6]. A large number of various plants such as seeds, pulps, fruits and plumules have been examined and cultivated as new oil crops. Niger seed (*Guizotia abyssinica*), Linseed (*Linumusitatissimum L.*), and Sesame (*Sesamumindicum L.*) are chief oil seed crops. However, interest in finding newer sources of edible oils has recently grown [3, 7, 8]. Globally, there is a growing demand for oil derived from edible oilseeds, particularly from indigenous seeds and plants, which are rich in essential nutrients and offer various medicinal benefits [9, 10]. Throughout history, people have utilized oils produced from oilseeds primarily for nourishment. Today, Indonesia and Malaysia are the top suppliers of palm oil, while Argentina, Brazil, and the USA dominate the soybean oil industry. The European Union and Ukraine are the largest suppliers of sunflower oil [3, 11].

In Ethiopia's case, oilseeds support the rural and national economy, produced by more than 3 million smallholders, and are the nation's second-biggest export earner after coffee. Niger seed is an oilseed cultivated in different parts of Ethiopia and India. It makes up roughly 50% and 3% of oilseed production in Ethiopia and India consecutively. The two nations are the world's top producers and consumers of Niger seeds [12]. It is also grown and used as a minor oil crop in certain African countries such as Sudan, Uganda, Kenya, and Malawi. In Ethiopia, it is grown on soggy soils, unsuitable for the growing of most other oilseeds or crops and contributes to soil conservation and land rehabilitation [13].

*Guizotia abyssinica* named Niger seed and locally known as "Noug", is used as a human food and good source of minerals and other nutrients for humans. It is very significant to Ethiopia's economy, providing between 50 and 60 percent of the country's edible oil supply. Its oil is also used in the manufacture of soaps and paints and as a lubricant or lighting fuel [14]. It is used as feed, fertilizer, or fuel. Particularly, in tropical regions, Niger seed meal plays a significant role as a supplement for sheep and goats due to the low quality of their baseline feed and the absence of locally sourced energy and protein sources [15, 16]. Hence searching

out the unique nature of the Niger seed used as a raw material in oil production is one part of promotion which has economically important for the large number of farmers of Ethiopia cultivating the Niger seed.

Report showed that polyphenols and phospholipids of different plant seed oil do have antioxidant action, which prevents lipid oxidation and enhance the shelf life of the oil material. The phospholipid compounds do their antioxidant action in collaboration with tocopherols, by forming an oxygen barrier between the oil-air interface, formation of melano-phospholipids, and in chelation of pro-oxidant metals. They also reported that glycolipids of plant seed oil had a strong antiradical activity due to its reducing sugars [8, 17]. However, the fatty acid types and composition, the concentration of other phytochemicals, including plant secondary metabolites, vary with respect to geographical origin, cultivation season and varietal or species type of the seed, which have a direct relation with the antioxidant activity of the edible seed or its oil product. To state it vividly, the mineral type and the climatic condition of the environment do direct influence on the type and amount of secondary metabolites (like the various phospholipids and polyphenols) derived from plants. Therefore, the determination of the fatty acids (saturated and unsaturated) composition, total phenolic compounds, total flavonoids, and its antioxidant activity in Ethiopian Niger seed (*Guizotia abyssinica*) is very important in assessing the standard and quality of Ethiopian Niger seed oil as well as any potential implications related to health.

## 2. Materials and method

### 2.1. Chemicals and reagents

All reagents and standards that were used in the analysis were of analytical grade. Standard grades of fatty acids were used for the laboratory analysis. Other chemicals are methanol (absolute acetone free), chloroform (99.99%), toluene (99.99%), chromatographic grade n-hexane with purity of 99.9%, anhydrous sodium sulfate (99%), sulfuric acid (98%), and acetone (99.5%). Analytical grade Sodium chloride. Aluminum chloride, Sodium nitrite, and undecanoic acid as internal standards were used for the laboratory analysis. Sodium tungstatedihydrate ($Na_2WO_4.2H_2O$), Catechin, Gallic acid, Ascorbic acid, and Sodium molybdate ($Na_2MoO_4.2H_2O$) were also used as standard chemicals. These chemicals and standards were obtained from Sigma Aldrich. Distilled and deionized water was used for solution preparation, dilution, and rinsing.

### 2.2. Apparatus and instruments

In addition to the common apparatuses available in any chemical laboratory (like test tube, beaker, round Bottle Flask, spatula), the major instruments and apparatuses used for our analysis were: Benchtop Shaker (ZHWY-304/334/344), GC-MS (Agilent Technologies 7890B-5977A) and Micro Pipette (from China), high speed Universal Disintegrator Electrical Girder (FW100), scientific natural flow type Oven (DAIHAN), Balance (RADWAG:ps360/c/1), Incubator (constant temperature and humidity incubator), Vacuum Rotary evaporator (Stone Staffordshire, England, ST 1S OSA), UV-vis (model 1601, Shimadzu, Kyoto, Japan), Portal centrifuge (Japan), Refrigerator (digital inverter technology, Samsung), and Crimper (crimper tool 11mm hand crimper, QTY:1) were employed throughout the analysis process. Plastic bags, aluminum foil, and Vials were also used in the analysis.

### 2.3. Sample collection

Samples of seed were collected from farmers in specific regions belonging to two prominent *G. Abyssinica* producers. Five zones from two different regions of Ethiopia were intentionally

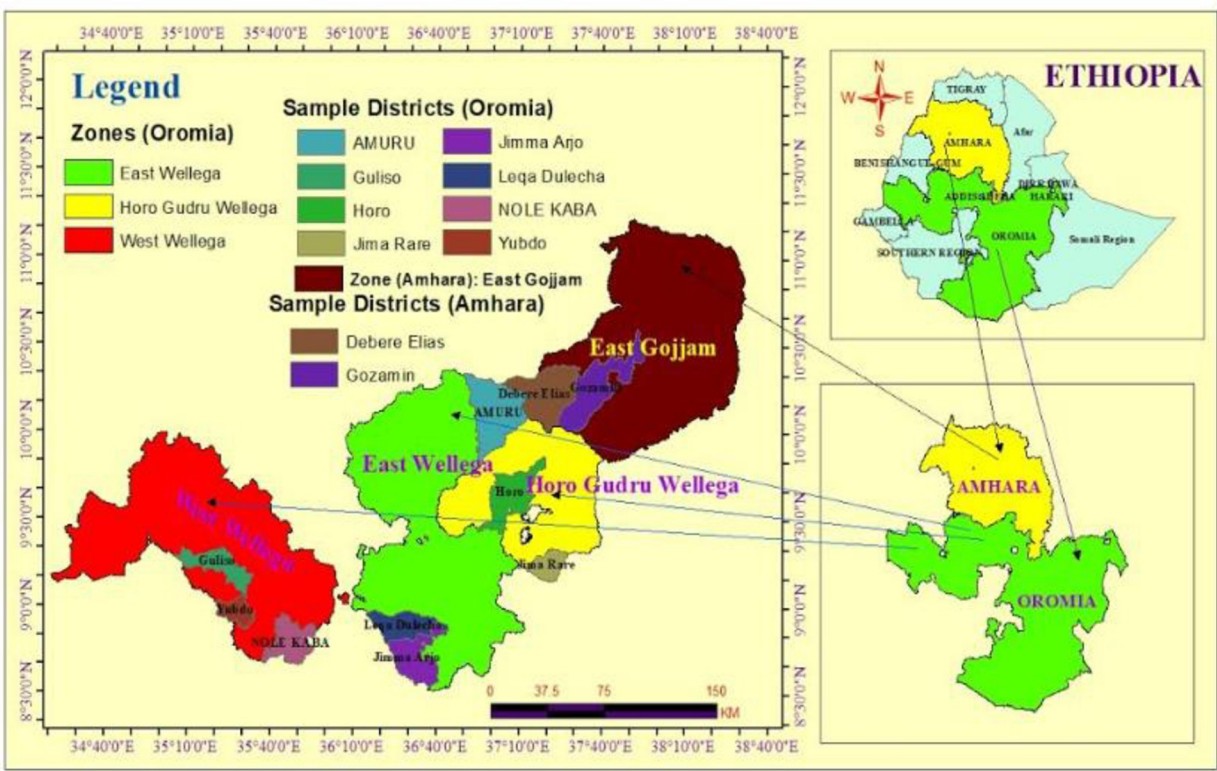

**Fig 1. The map showing the selected five districts of the Oromia and Amhara (source: EMA data using ArcGIS 10.3 software).**

selected as sampling areas. From the Oromia region, three zones were selected: Horo Guduru Wolega zone (HG/Wolega), East Wollega zone (E/Wolega), and West Wolega zone (W/Wolega). The North Gondar zone (N/Gondar) and East Gojjam Zone (E/Gojjam) were chosen from the Amhara region. For each of the five zones, two to three districts were carefully chosen to collect the samples: From Horo Guduru Welega zone (Amuru and Jimma Rare districts), from East Wollega zone (Leka Dulecha and Jimma Arjo districts), from West Wollega zone (Yubdo, Nolekaba, and Guliso district), from North Gondar zone (Lay Armachiho and mirab Armachiho district), from East Gojjam Zone (Guzamn and Debre Elias districts) were considered for sampling. Fig 1 shows the map of the sampling regions, zones, and districts. Since 1988, Ethiopia has released four indigenous varieties of niger seed: Fogera and Esete-1 (both introduced in 1988), Kuyu (introduced in 1994), Shambu-1 (PGRC/E 228423, released in 2002), and Ginchi (released in 2010). The Ginchi varieties are the most prolific and extensively farmed in the country and have been considered in this study. After being cleaned of dust particles with distilled water, the dried seed samples were allowed to dehydrate at room temperature. Prior to extraction, the dehydrated materials were finely blended in an electric blender, sieved through a 0.25–50 mm screen, and sealed in an airtight container.

## 2.4. Extraction procedure

**2.4.1. Extraction of lipids.** Lipids from niger seeds were extracted following standard methods with only a few modifications [18]. In this method, 0.50 g of niger seed sample powder was extracted with a mixture of chloroform and methanol 12.00 mL in 2:1 volume ratio, respectively. The filtrate was taken by centrifuging the extract after shaking for 36 hours on a

platform shaker (at 300 rpm). In the filtrate, 2.00 mL of 0.73% aqueous sodium chloride was introduced, and the upper phase was removed by siphoning process using a micropipette while the lower phase (chloroform layer) containing the lipids was recovered. Eventually, the solvent was removed using a rotary evaporator under vacuum and the residue was redissolved with 5.00 mL of toluene and kept for analysis.

**2.4.2. Derivatization of fatty acids.** The polar carbonyl groups must first be changed to more volatile non-polar derivatives in order to allow GC analysis [19]. About 1.00 mL of the lipid extract was taken from the redissolved lipid extract keep for analysis, and 175.00 µL, 0.1697 mg/mL of unidecanoic acid was spiked and allowed to react with 2.00 mL of 1% methanolic sulfuric acid solution for 12 hours by keeping the reaction temperature at about 50˚C in an incubator. Then after, the mixture was treated with 5.00 mL, 5% sodium chloride aqueous solution. After phase separation, the required ester was extracted with hexane (2×3 mL) and the upper phase was taken away by using a micropipette (siphoning). The derivatized molecule was allowed to dry over anhydrous sodium sulfate, filtered by an acrodisc membrane syringe, transferred into the vial and analyzed by GC-MS.

**2.4.3. GC-MS analysis.** A gas chromatograph instrument hyphenated to an electron ionization mode-operated mass spectrometer was used for analysis. The GC separation was performed using a fused-silica chemically bonded phase capillarity column having 30 m, 0.25 mm I.D., 0.25 mm film thickness. The opening oven temperature was 125˚C, then after 2 min it was allowed to step up to 170˚C at the rate of 30˚C per minute. By changing the rate to 15˚C/min, it was upstretched to 200˚C. After keeping it for 2 min, it was allowed to raise to 230˚C with a frequency of 3˚C per minute, and it was detained for 20 min. The other operating conditions of the GC system were injector temperature, 250˚C with a split ratio of 25:1 and the carrier gas is He (99.999%) at a flow rate of 1 ml/min. The temperature of the ion source is 230˚C, the quadruple temperature is 150˚C, and the solvent delay time is 3.5 min. Eventually, the spectra was documented at 70 eV with a mass range from *m/z* 40 to 500 amu.

**2.4.4. Quantification of fatty acids.** The amount of all sensed fatty acids having relative percentages higher than 0.1%, were quantified according to Eq 1 proposed by Dussert and his cowriters [20].

$$\frac{W}{W}\left(^{mg}/_{g}\right) = \frac{AFA \times MIS}{AIS \times MC} \tag{1}$$

Where, AFA *is the peak area the fatty acid*, **AIS** *is the internal standard peak area*, **MIS** *is the mass of the internal standard and* **MC** *is the mass of niger seed used for the analysis.*

The total fatty acid concentration was determined by summing up individual fatty acid concentration.

$$TFA = \sum FAi \tag{2}$$

Where, **TFA** *is total fatty acid concentration* and **FAi** *is individual fatty acid concentration.*

## 2.5. Experimental procedures

**2.5.1. Preparation of crude *G. Abyssinica* extracts.** The ultrasound assisted extraction procedure, presented by Goli et.al. was used for sample extraction with few modifications [21]. Briefly, 1.00 g of powdered niger seed sample was taken and transferred into 25 mL separate plastic test tube. 25.00 mL of 100% methanol and 80% aqueous methanol was introduced into each test tube. Then, the sample was extracted in an ultrasonic bath for 60 min at 30˚C and centrifuged. Then the supernatant was taken after filtering through the Whatman number one

filter paper into 25 mL test tube. Finally, the test tube was filled up to the mark with respective solvents.

**2.5.2. Total polyphenol determination.** The total polyphenol compounds content was estimated using the standard colorimetric methods with very few modifications [22]. In this method, exactly 0.10 mL of the seed sample extracts (0.667 g/ml) were mixed with 1.50 mL of double distilled water, 0.75 mL of Folin-Denis reagent and 3.75 ml of 7% sodium carbonate. After that, the mixture was allowed to sit at ambient temperature in the dark for 30 minutes, and the absorbance of the mixture was measured at 760 nm using a spectrophotometer. The results were expressed in milli grams of gallic acid equivalent per gram of dry sample (mgGAE/g of dry sample).

**2.5.3. Total flavonoids content determination.** A standard colorimetric method with the use of Aluminum chloride was applied to estimate the total flavonoid content with some modifications [23, 24]. The reaction mixture (6.3 mL) comprised of 0.2 mL of extracted sample, 3.5 mL distilled water and 0.3 mL of sodium nitrite (5%) was added, then after 5-minute 0.3 mL of Aluminum chloride (10%) was added; again, after extra 5-minute 2 mL of 4% sodium hydroxide was mixed. The mixture was incubated at room temperature for 30 min prior to measuring of the absorbance at 510 nm. The standard curve was generated using catechin standard solutions following the same fashion as the sample and the amount of total flavonoids content was documented in terms of milli gram of Catechin equivalent (mgCE/g of dry sample).

**2.5.4. Measure of antioxidant activities.** *2.5.4.1. DPPH radical scavenging activity determination*. A standard method was selected for the antiradical properties of the crude niger seed oils under study. The antioxidant potential of the samples were predicted by comparing the scavenging power of the sample with ascorbic acid standard towards the stable free radicals DPPH, based on the standard methods [8, 25]. Seven various volumes of both 100% and 80% methanol extracts (5, 10, 20, 40, 50, 75 and 100 μL) were added into separate test tubes. Then, 2mL of 0.004% solution of DPPH radical solution in methanol was added to each test tube with a vortex mixer. Finally, the test tubes were filled with 100% methanol and 80% methanol to the mark (5mL). The solution mixture was incubated for 30 minutes in the dark at room temperature. The scavenging capacity of the sample was studied by monitoring the decrease in absorbance of the mixture at 517 nm spectrophotometrically. Finally, inhibition of free radical DPPH in percentage (%) was calculated using Eq 3

$$\% \ Inhibition = \frac{(A_o - A_s)}{A_o} * 100 \tag{3}$$

Where **Ao**—is *the absorbance of the control (blank) sample and* **As**—is *the absorbance of the various extract samples.*

$IC_{50\%}$ value of the extracts, reported as μg/mL, denotes the concentration of crude extract in DPPH methanol solution needed to cause 50% inhibition of DPPH radicals. A graph was plotted for percent DPPH leftover against the concentration of oil in DPPH methanol solution (mg/mL), and $IC_{50}$ value was determined by interpolation of values on the graph.

## 2.6. Data analysis

In this study, results for each test parameter were interpreted and statistical analysis was done by using SPSS version 20, Origin 6.0 and Excel 2007 software. For fatty acid analysis, duplicate analysis was done, while for other parameters a triplicate analysis was made on each sample, and standard statistical methods were used to calculate the mean values and standard deviations. Microsoft Excel spreadsheet was used to process and reorganize the data. One-way analysis of variance (ANOVA) and Tukey was used to compare means among treatments.

**Table 1. Names, retention times (RT), and mean (n = 2) relative percentage values of the identified FAs in the Niger seeds samples taken from five zones.**

| No | Chemical Name | FAs | Mean values (%) of identified FAs in Niger Seeds sites | | | | | RT |
|---|---|---|---|---|---|---|---|---|
| | | | W/Wolega | E/Wolega | HG/Wolega | N/Gondar | E/Gojjam | |
| 1 | Methyl tetradecanoate | C14:0 | 0.070 | 0.058 | 0.049 | 0.064 | 0.0606 | 7.257 |
| 2 | Tetradecanoic acid, 12-methyl-, methyl ester, (S)- | - | 0.024 | 0.0175 | 0.016 | 0.0176 | 0.015 | 8.438 |
| 3 | 7-Hexadecenoic acid, methyl ester, (Z)- | C16:1(n-9) | 0.146 | 0.142 | 0.114 | 0.155 | 0.106 | 9.566 |
| 4 | Hexadecanoic acid, | C16:0 | 10.391 | 10.576 | 10.317 | 10.662 | 10.536 | 9.883 |
| 5 | 9,12-Octadecadienoic acid (Z,Z)-, methyl ester | C 18:2(n-6) | 72.887 | 67.692 | 74.669 | 67.302 | 71.128 | 12.923 |
| 6 | 11-Octadecenoic acid, methyl ester | C 18:1(n-7) | 0.448 | 0.453 | 0.429 | 0.462 | 0.465 | 13.018 |
| 7 | 9-Octadecenoic acid (Z)-, methyl ester | C18:1(n-9) | 5.769 | 10.909 | 5.438 | 11.017 | 7.696 | 13.119 |
| 8 | Methyl stearate | C18:0 | 8.761 | 8.542 | 7.59 | 8.703 | 8.541 | 13.527 |
| 9 | 17-Octadecynoic acid (17-ODYA) | - | 0.0304 | - | 0.073 | - | - | 17.281 |
| 10 | Eicosanoic acid, methyl ester | C20:0 | 0.532 | 0.527 | 0.427 | 0.064 | 0.499 | 17.895 |
| 11 | Docosanoic acid, methyl ester | C 22:0 | 0.588 | 0.682 | 0.606 | 0.0176 | 0.577 | 24.165 |
| 12 | Tetracosanoic acid, methyl ester | C24:0 | 0.352 | 0.399 | 0.273 | 0.155 | 0.311 | 35.174 |

Significant differences were determined at P < 0.05 levels, and results were expressed as mean ± standard deviation (SD).

## 3. Results and discussions

### 3.1. Qualitative identification of fatty acids in the samples

As can be seen from Table 1 and Fig 2, the major fatty acids in Niger seed are represented by unsaturated FAs. The principal unsaturated fatty acids in all Niger seed samples are Linoleic

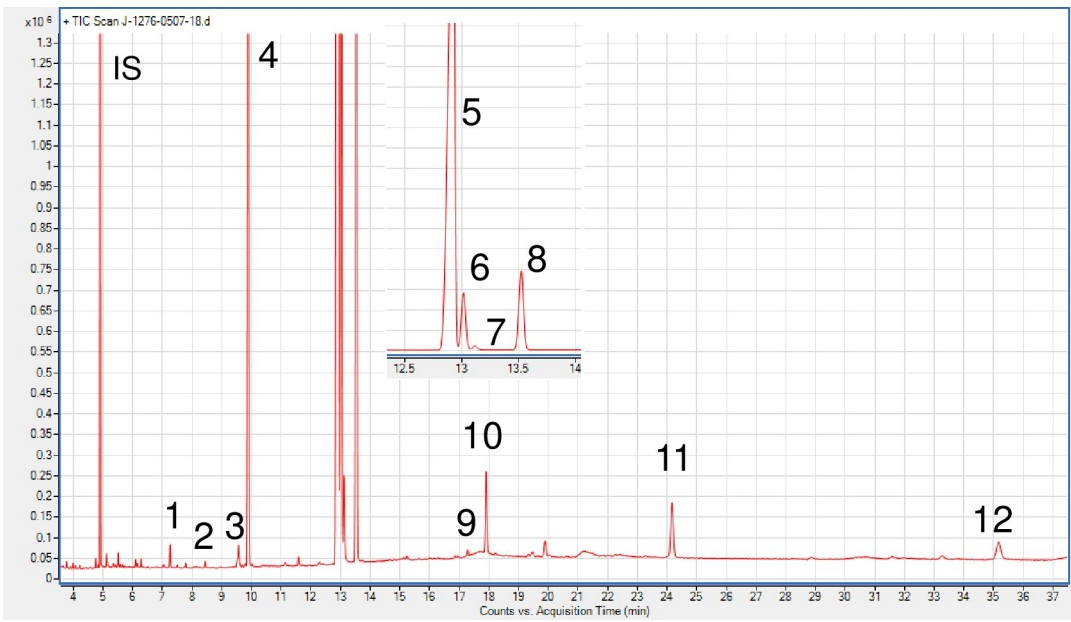

**Fig 2. GC-MS chromatogram for qualitative identification of fatty acids investigated in niger seed sample. (1.** *Methyl tetradecanoate;* **2.** *Tetradecanoic acid, 12-methyl-, methyl ester, (S)-;* **3.** *7-Hexadecenoic acid, methyl ester, (Z)-;* **4.** *Hexadecanoic acid;* **5.** *9,12-Octadecadienoic acid (Z,Z)-, methyl ester;* **6.** *11-Octadecenoic acid, methyl ester;* **7.** *9-Octadecenoic acid (Z)-, methyl ester;* **8.** *Methyl stearate;* **9.** *17-Octadecynoic acid (17-ODYA);* **10.** *Eicosanoic acid, methyl ester;* **11.** *Docosanoic acid, methyl ester;* **12.** *Tetracosanoic acid, methyl ester*).

acid C18:2(n-6); ranging from 67.30 to 74.67%, taking the first rank followed by Oleic acid C18:1(n-9) having values from 5.44–11.02% and vaccenic acid C18:1(n-7) with mean percentage values from 0.43–0.47%, irrespective of variety and location.

Palmitic (C16:0) and stearic (C18:0) acids were the major saturated fatty acids in the five Niger seeds samples accounting for 10.32 to 10.66% and 7.59–8.76%, respectively (Table 1). This range is almost in agreement with the data reported by Dutta and his cowriters on Ethiopian niger seeds varieties. In their report, the concentration of Linoleic acid (C18:2) ranged from 71.4 to79.2%, Oleic (C18:0) at a range of 6 to 11% each, Palmitic (C16:0) and stearic (C18:0) acids ranged from 9.4 to 16.6% and 6.6 to 8.2% respectively [26].

## 3.2. Relative percentage composition and actual concentrations of Individual Fatty acids and total fatty acid concentration in the samples

The concentrations (w/w) of the different individual fatty acids (FAs) in Niger seed *(Guizotia abyssinica)* samples were quantified and documented relative to the internal standard (Undecanoic acid) using Eq (1) as shown in Table 2. The total concentration of FAs in Niger seed (*Guizotia abyssinica*) is the sum of each individual fatty acid concentration (saturated and unsaturated fatty acid) was calculated using Eq 2 and documented as tabulated in Table 2.

The fatty acid content of Niger seed samples taken from Amhara region (East Gojjam and North Gondar Zones) and Oromia Region (East Wolega, West Wolega, and HG/Wolega) zones reveals considerable differences in some of the fatty acids, especially the unsaturated fatty acids. The results disclose that Niger seed oil is predominantly constituted of unsaturated fatty acids, particularly linoleic acid. This finding corroborates with earlier studies reported in *G. Abbysinica* samples [13]. Although most of the identified fatty acids show minor variability, regional and zonal variations were identified in the level of some fatty acids, such as vaccenic acid and oleic acid.

The saturated fatty acids were mainly palmitic and stearic acids, with the highest concentration of palmitic acid in N/ Gondar and East Wolega, 37.1 mg/g and 30.9 mg/g, respectively. Stearic acid also contributes to the total amount of saturated fat, standing at 26.7 mg/g in East Gojjam. Generally, the percentage of saturated fatty acids is normally low, hence indicating

**Table 2. The concentration of FAs determined in Niger seed samples (mg/g dry weight).**

| Name of the FAMEs | Common name of the FAs | Concentration of FAs (*mg/g*) in samples from five zones | | | | |
|---|---|---|---|---|---|---|
| | | W/Wolega | E/Wolega | HG/Wolega | N/Gondar | E/Gojjam |
| Methyl tetradecanoate | Myristic acid | 0.2 | 0.17 | 0.12 | 0.22 | 0.19 |
| Tetradecanoic acid, 12-methyl-, methyl ester, (S)- | 12-methyl myristic acid | 0.07 | 0.05 | 0.04 | 0.06 | 0.05 |
| 7-Hexadecenoic acid, methyl ester, (Z)- | cis-Hypogeic Acid | 0.41 | 0.42 | 0.28 | 0.54 | 0.33 |
| Hexadecanoic acid, methyl ester | Palmitic acid | 29.1 | 30.9 | 24.8 | 37.1 | 32.9 |
| 9,12-Octadecadienoic acid (Z,Z)-, methyl ester | Linoleic acid | 204 | 198 | 179 | 234 | 222 |
| 11-Octadecenoic acid, methyl ester | Vaccenic acid | 16.2 | 31.9 | 13.1 | 38.3 | 24.1 |
| 9-Octadecenoic acid (Z)-, methyl ester | Oleic acid | 1.26 | 1.32 | 1.03 | 1.6 | 1.46 |
| Methyl stearate | Stearic acid | 24.6 | 25 | 18.3 | 30.3 | 26.7 |
| 17-Octadecynoic acid | Alkynyl Stearic Acid | 0.09 | Trace | 0.17 | Trace | Trace |
| Eicosanoic acid, methyl ester | Arachidic acid | 1.49 | 1.54 | 1.03 | 1.83 | 1.56 |
| Docosanoic acid, methyl ester | Behenic acid | 1.65 | 1.99 | 1.46 | 2.38 | 1.8 |
| Tetracosanoic acid, methyl ester | Lignoceric acid | 0.99 | 1.17 | 0.66 | 1.41 | 0.97 |
| **Sum Total** | | **280.06** | **292.46** | **239.99** | **347.74** | **312.06** |
| Saturate Fatty acid (SFA) | | 58.03 | 60.77 | 46.37 | 73.24 | 64.12 |
| Unsaturated Fatty acid(UFA) | | 222.03 | 231.69 | 193.62 | 274.5 | 247.94 |

that Niger seeds are not rich in saturated fat. The variability of the two dominant saturated fatty acids is insignificant in samples of five zones and the two regions.

The unsaturated fatty acids are highly abundant (Table 2), and they are accredited with a number of health beneficial qualities. In this regard, oleic acid is at 1.6 mg/g in N/Gondar, a monounsaturated fatty acid associated with a reduction in the risk of heart disease. Linoleic acid, on the other hand, is an omega-6 fatty acid, well represented in all samples, with the most representation in N/Gondar (234 mg/g). This is considered as an essential fatty acid because of its importance to cell function, especially in inflammation processes. Considering the sum of all the identified fatty acid content, the samples taken from the Amhara Region, (from North Gondar and East Gojjam's zones), are substantially higher, with North Gondar exhibiting the highest total at 347.74 mg/g followed by East Gojjam at 312.06 mg/g. On the other hand, samples from the three zones belonging to the Oromia region showed lower total fatty acid content, as compared to the Amhara region's sample, where the total fatty acid values ranged nearly from 240 mg/g in a sample from Horo Guduru Wolega to 292.46 mg/g in the sample taken from East Wolega zone. In the Amhara Region, the greater total fatty acid contents suggest a robust nutritional profile, particularly with respect to beneficial fatty acids such as oleic acid and vaccenic acid.

The total saturated fatty acids fluctuated between 46 and 73 mg/g, while unsaturated fatty acids spanned from 194 to 275 mg/g. In comparison to the total fatty acids, the N/Gomdar sample exhibits a higher concentration of unsaturated fatty acids, succeeded by HG/Wolega (Table 2). This suggests that niger seed is nutritionally superior compared to other oilseeds, which contain considerably greater unsaturated fatty acids. Oil rich in unsaturated fatty acids is considered more beneficial for eating than oil high in saturated fatty acids.

For instance, N/Gondar has 38.3 mg/g of vaccenic acid, which could give anti-inflammatory benefits, whereas oleic acid reaches 1.6 mg/g. These findings indicate that Niger seeds from the Amhara Region may be particularly helpful for health-oriented uses.

The observed variations in fatty acid content can be related to numerous factors, including soil type, climate, and specific agricultural practices practiced in each region. The varied agroecological circumstances in Oromia and Amhara presumably influence the metabolic pathways in Niger seeds, influencing their fatty acid compositions.

## 3.3. Total phenolic content (TPC) and Total flavonoid contents (TFC)

In this study, the concentration of total phenolic content and total flavonoid content of the samples studied were determined from the calibration curve constructed using gallic acid (Fig 3) and Catechin (Fig 4), respectively. A good regression coefficient ($R^2$ >0.99) was obtained for the two classes of compounds, suggesting a good correlation between the concentration of the standard solution and absorbance (Figs 3 & 4). The total phenolic content (TPC) was expressed as per milligram gallic acid equivalent per gram of sample (mg GAE/g), and the total flavonoids content (TFC) was expressed as milligram of Catechin equivalent per gram of sample (mg CE/g). The results are summarized in Table 3.

Table 3 displays the TPC and TFC of Niger seed samples taken from the five zones (East Wolega, West Wolega, HG/Wolega, N/Gondar, and Esast Gojjam) where,.the former three zones belong to the Oromia region, and the other two belong to the Amhara region. The effect of extracting solvent (methanol and aqueous methanol) was also compared, and results are presented in Table 3. Considering the table, the TPC content of Niger seed samples exhibited significant variation among the various zones examined. The maximum TPC value was observed in the Niger seed sample taken from the North Gondar zone (Amhara region) at 11.78 mg GAE/g in 80% methanol, and East Gojjam at 11.58 mg GAE/g which are statistically

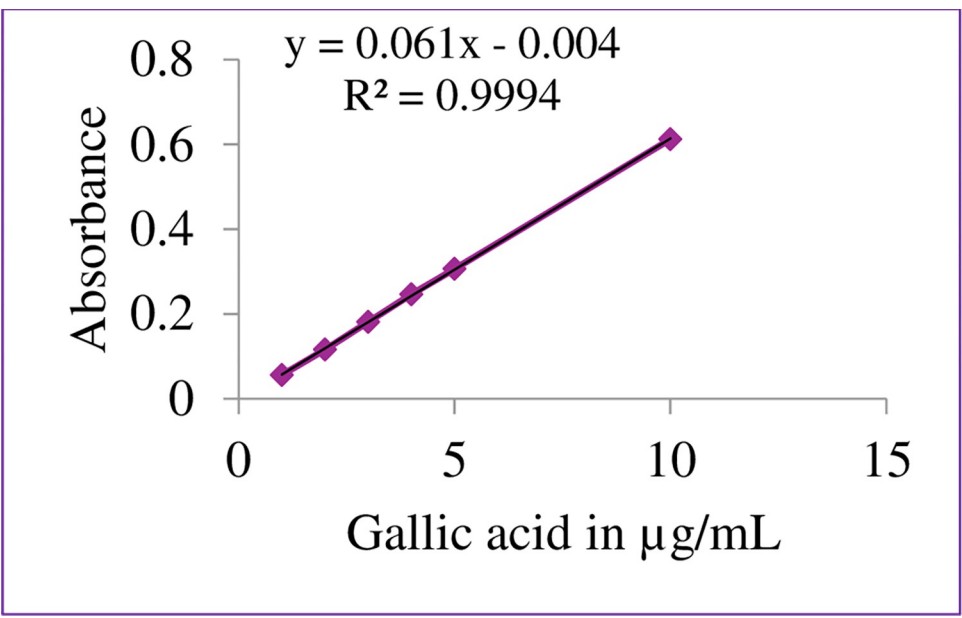

**Fig 3. Gallic acid calibration curve for TPC.**

similar. In general, the samples taken from the Amhara region typically include greater phenolic content than those from the Oromia region, where the maximum value recorded in East Wolega was 11.49 mg GAE/g using 80% methanol. The minimum TPC was recorded in West Wolega, with measurements of 10.89 mg GAE/g in 80% methanol and 7.65 mg GAE/g in 100% methanol, indicating regional and zonal variations in the phenolic composition of Niger seeds.

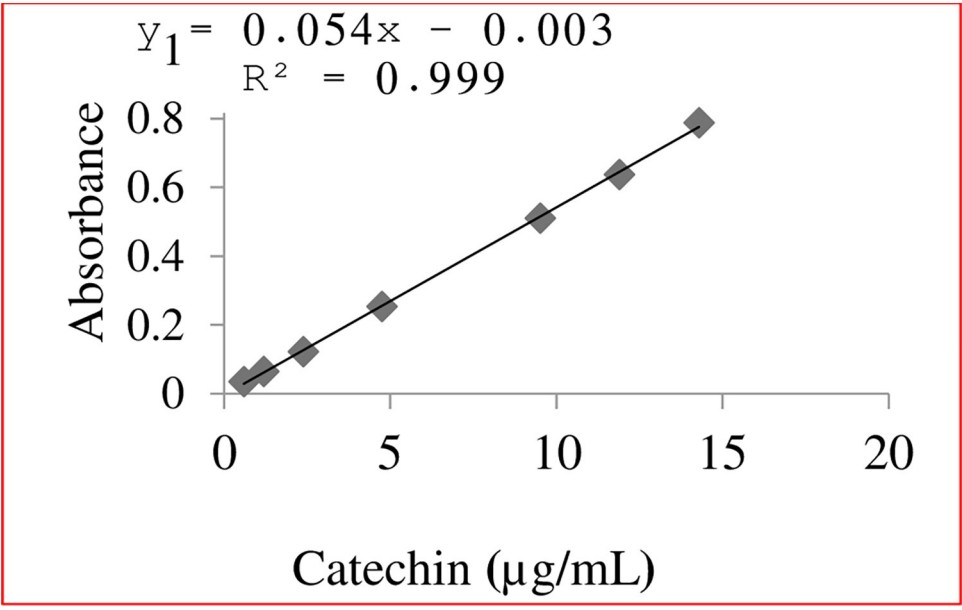

**Fig 4. Catechin calibration curve for TFC.**

**Table 3. Total phenolics, and total flavonoids contents of *G. Abyssinica* seed sample extract using methanol.** (mean ± SD, n = 3).

| Sample | Total phenolic Contents (mg GAE/g) | | Total flavonoid (mgCE/g) | |
|---|---|---|---|---|
| site | 80% Methanol extract | 100% Methanol extract | 80% Methanol extract | 100% Methanol extract |
| W/Wolega | 10.89±0.39[c] | 7.65±0.38[d] | 5.64±0.09[b] | 5.00±0.11[d] |
| E/Wolega | 11.49±0.15[ab] | 9.96±0.23[bc] | 6.44±0.02[d] | 5.87±0.01[c] |
| HG/Wolega | 11.24±0.08[b] | 10.92±0.38[a] | 6.04±0.22[c] | 6.31±0.22[b] |
| N/Gondar | 11.78±0.028[a] | 9.56±0.09[c] | 6.67±0.06[e] | 4.98±0.03[d] |
| E/Gojjam | 11.58±0.04[a] | 10.28±0.52[b] | 5.42±0.03[a] | 6.76±0.22[a] |

Note: Means with common letters (a-e) within a column are not statistically significantly different (p ≥ 0.05)

In solvent comparisons, 80% of methanol consistently extracted more phenolics than 100% in all zones. In HG/Wolega, the total phenolic content (TPC) with 80% methanol was 11.24 mg GAE/g, whereas it decreased to 10.92 mg GAE/g with 100% methanol. This pattern was also observed in samples of the other region, confirming that aqueous methanol is a superior solvent for extracting phenolic chemicals. The higher effectiveness of 80% methanol may be owing to its capacity to dissolve a broader spectrum of chemicals, including phenolics, which are often more soluble in polar solvents, especially simple phenolics. The findings of this study was partly in line with the study reported by [27].who reported that the 70% methanol extract from white mustard had the highest phenolic content, followed by rapeseed and camelinain. And also, the presented study was in agreement with the work of Teh et al, (2014), [28] where TPC, Antioxidant activity and TFC of 80%methanol extract seed cakes shows highest value than 100% methanol extract. On the other hand, 100% methanol was found to be efficient as compared to 80% and 50% methanol for extraction of phenolic compounds from *Kirkia wilmsii uber*, [29]. This suggests that the efficiency of solvent composition might be dependent on the nature of phenolic compounds due to the fact that it varies from plant to plant. For example, the polymerized phenolic such as tannins are more soluble in medium polar solvents as opposed to simple phenolic compounds [30].

The total flavonoid content of the samples taken from the five zones is depicted in Table 3. Looking at the Table, the TFC of the Niger seed samples, like TPC, also showed a significant variation among the different zones. The highest TFC value was recorded by the East Gojjam (Amhara region) samples in 100% methanol at 6.76 mg CE/g, whereas the North Gondar samples at 80% methanol had a higher TFC, 6.67 mg CE/g. Once again, flavonoid content in seeds appears to be higher in samples from the Amhara region, consistent with phenolic content. The lowest TFC in seeds was again from West Wolega, 5.00 mg CE/g (100% methanol) and 5.64 mg CE/g (80% methanol). These findings again demonstrated the variation in the phytochemicals of the seeds between zones. Generally, 80% methanol was a better solvent for extracting flavonoids from whole Niger seed samples than 100% methanol: East Wolega samples had TFCs of 6.44 mg CE/g and 5.87 mg CE/g for 80% and 100% methanol extract, respectively. For East Gojjam, however, 100% methanol was better than 80%, the values being 6.76 mg CE/g and 5.42 mg CE/g for 100% and 80% methanol, respectively. Clearly, for some zones, the less polar 100% methanol extracted more flavonoids. This may be due to the types and natures of flavonoids in the Niger seeds differing between zones, such that the flavonoids in East Gojjam are soluble in non-polar solvents. Comparing region wise, samples from Amhara region zones, contained relatively higher TPC as well as TFC values than the Oromia zones particularly in 80% methanol extract. Statistical analysis of the data showed that the difference between the zones was significant (p < 0.05), indicating that the zone of collection has a major

effect on the TFC of Niger seeds. The higher values for flavonoid content in East Gojjam and North Gondar zones could be again due to differences in soil nutrient type and availability or the climate of these areas. These environmental factors may influence the biosynthesis and accumulation of flavonoids in the seeds. It has been reported that the types of flavonoids, their concentrations, and solubility varies with respect to the species of the samples under investigation, region of origin, and varietal or accession types of the samples under investigation [30–32].

### 3.4. Antioxidant activity of the Niger Sample using DPPH scavenging assay

The free radical scavenging capacity of the Niger seed extract taken from 100% methanol and 80% aqueous methanol extract was assayed. The DPPH radical scavenging activities of the seed extracts were determined by comparing the scavenging activity percentage value of the DPPH with a standard ascorbic acid (Fig 5).

Up on increasing extract concentration, the percent inhibition also increased. The present study showed that the ability of Niger seed of 100% methanol extracts to scavenge the DPPH radical was at the level ranging from 0.91–92.15%, and for 80% aqueous methanol extract, it was at the level ranging from 12.91–94.53% (Fig 6). In the presented study, 80% methanol extract showed the highest percent inhibition in all Niger seeds extracts.

### 3.5. The $IC_{50}$ values of each sample

The $IC_{50}$ value of 80% aqueous methanol extract was lower when compared with 100% methanol extract, as can be seen from Table 4. The lower the $IC_{50}$ value, the higher the antioxidant activities of the tested sample, and the higher the $IC_{50}$ value, the lower the antioxidant activities. The niger seed sample extracted from 80% aqueous methanol and 100% methanol, taken from East Gojjam and HG/Wolega, scored lower $IC_{50}$ values: 132.79 ± 3.17 and 258.72 ± 2.75 µg/mL, respectively. However, the samples from East Wolega scored higher $IC_{50}$ values, 190.50 ± 1.46 and 378.30 ± 2.64, taken from both 80% aqueous methanol and 100% methanol extract, respectively (Table 4).

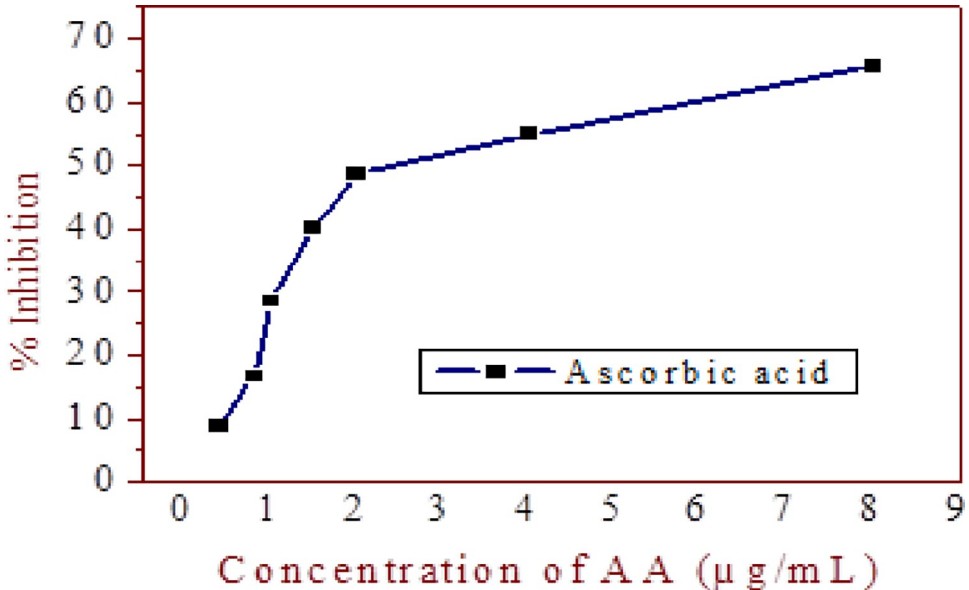

**Fig 5. Radical scavenging potential of the standard ascorbic acid (AA) for determination of antioxidant activity (AOA)of the niger sample.**

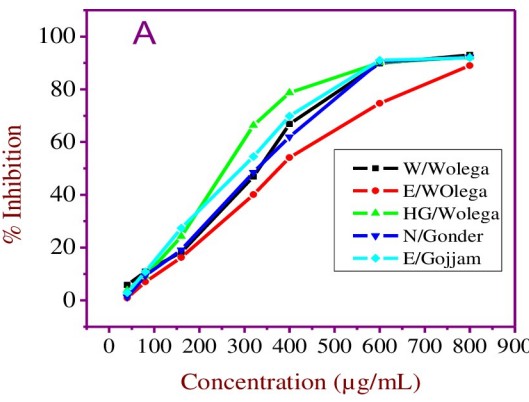
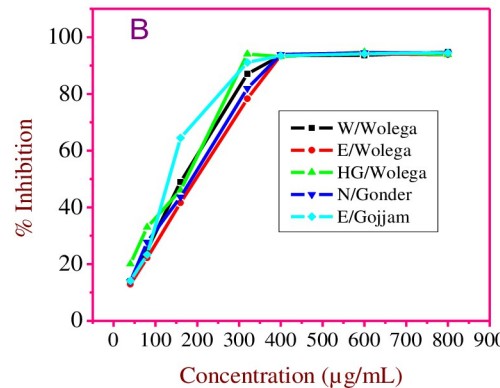

**Fig 6.** Free radical scavenging of Niger seed A) 100% methanolic extracts and B) 80% methanolic extract.

### 3.6. Correlation between TPC, TFC and antioxidant activities

The relationship between antioxidant activities and the total phenolic content of five niger seeds of 100% methanol and 80% aqueous methanol extracts was examined by plotting total phenolic contents versus the $IC_{50}$ values, as indicated in Fig 7. Antioxidant activity for 80% methanol and 100% methanol extracts of the niger seed had poor correlation with total poly-phenol content. The correlation coefficient ($R^2$) values for each of them were 0.079 and -0.072 respectively. The poor correlation indicates that in addition to the phenolic compounds, there were other constituents, like the fatty acids, which were involved in the scavenging effect of the seed. Although most reported studies indicate the presence of strong positive correlation between TPC and antioxidant activities, some studies show the existence of poor and negative correlation with the aforementioned parameters that support our study [33, 34]

Furthermore, an analysis was conducted on the correlation between the total flavonoid contents in *G. Abyssinica* and the scavenging potential of the samples at each site. The findings revealed a negative linear correlation (R = -0.5) for the 100% methanol extract and a substantial positive linear correlation (R = 0.895) for the total flavonoids and the radical scavenging activity of the 80% methanol extract (Fig 8). This finding indicates that 80% of methanol solution is known to be effective and the solvent system is widely applicable to extract natural anti-oxidative components, especially the flavonoids among phenolic compounds from plant materials. One possible reason for this is that the methanol-water mixture possessing high polarity and thus greater efficacy towards the extraction of polar phytochemicals such as poly-phenols and flavonoids [35].

**Table 4.** $IC_{50}$ values of Niger seed extract in 80% and 100% methanol (Mean ± SD, n = 3).

| Sample Site | DPPH Scavenging $IC_{50}$ value (µg/mL) from | |
|---|---|---|
| | *80% aqueous methanol* | *100% methanol extract* |
| W/Wolega | 167.56 ± 3.76[b] | 333.18 ± 2.64[c] |
| E/Wolega | 190.50 ± 1.46[d] | 378.30 ± 2.64[d] |
| HG/Wolega | 174.45 ± 3.11[c] | 258.72 ± 2.75[a] |
| N/Gondar | 188.02 ± 3.75[d] | 330.57 ± 2.64[c] |
| E/Gojjam | 132.79 ± 3.17[a] | 294.18 ± 4.08[b] |
| Ascorbic acid | 2.49 ± 0.08 | — |

Note: Mean values with common superscript letters (from a–d) within a column are not statistically significantly different (p ≥ 0.05).

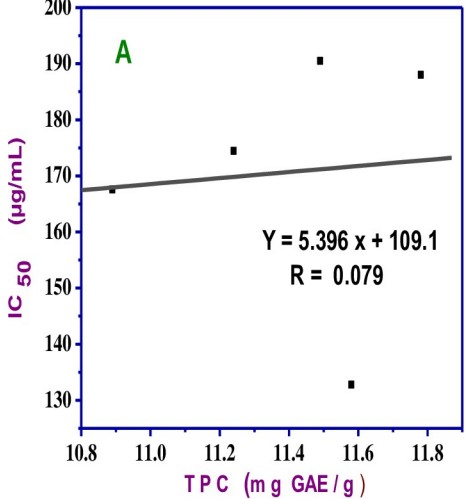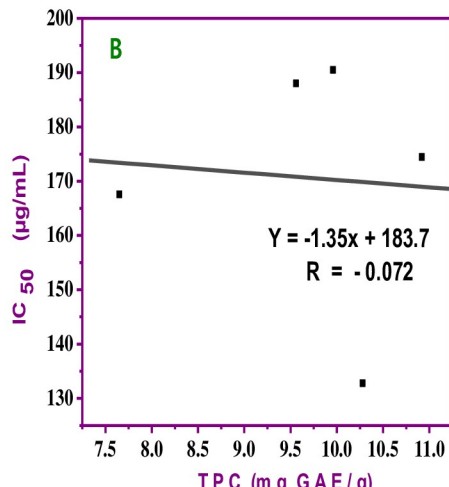

**Fig 7.** Correlation.between total polyphenol content and antioxidant potentials of A) 80% aqueous methanol and B) 100% methanol extracts of niger seed sample.

While there are other plant components with antioxidant action as well, polyphenols are the major classes of compounds present in most plants [36]. The relationship between TPC and the antioxidant activity of some extracts may also be hampered by certain nonphenolic chemicals (d-glucose, ferrous sulfate, and citric acid) that are known to react with Folin-Cio-calteu reagent but are ineffective as free radical scavengers [37]. Furthermore, several reports indicated that only certain hydroxyl positions of the phenolic and flavonoids group are responsible for determining the antioxidant activity of the molecules, and hence positive and strong correlation may not be expected for all samples and solvent extracts [34].

## 4. Conclusions

The present work, therefore, reports the detailed analysis of the fatty acid composition, TPC, TFC, and antioxidant activity (AA) assay of Niger seeds (*G. abyssinica*) obtained from five

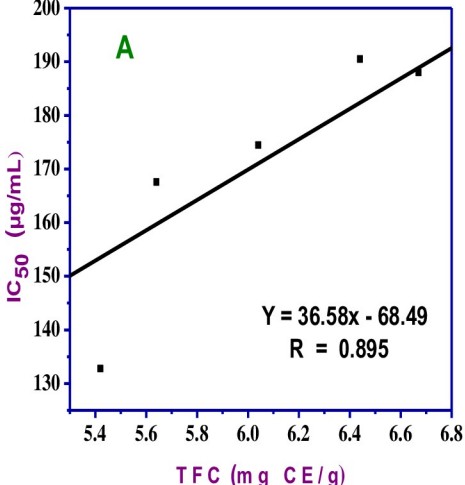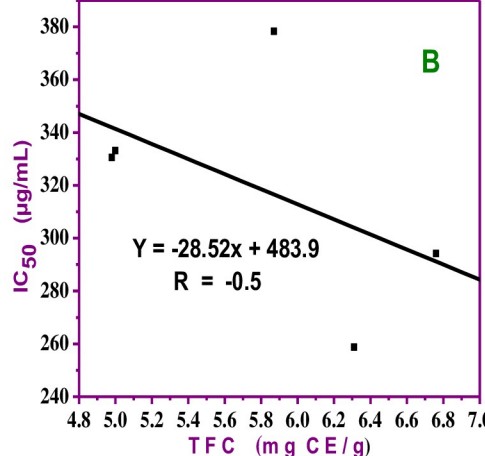

**Fig 8.** Correlation between total flavonoid content and antioxidant actions of A) 80% methanol and B) 100% methanol sample extract.

different zones in Ethiopia, which exhibited marked variation in both the quality and quantity of the fatty acids. Linoleic acid was the major fatty acid, constituting 67–74.7% of the total fatty acids, with N/Gondar having the highest amount of 74.7%. The share of the saturated fatty acids was also high; the palmitic and stearic acids were dominating, while the value of total fatty acids was highest in samples of N/Gondar zone (Amhra region) as 347.74 mg/g. These result indicates that the geographical areas have an influence on the fatty acid profile, hence HG Wolega had low contents of fatty acid 239.99 mg/g and unsaturated fatty acids.

Moreover, TPC and TFC were varied over zones. Accordingly, the highest values of TPC and TFC were recorded for samples from N/Gondar and E/Gojjam zones, respectively. The poor correlation between TPC and AA confirms that other constituents, such as fatty acids, might contribute to the AA of the seeds. These findings highlight the fact that Niger seed could be a rich source of essential fatty acids and bioactive compounds with implications for dietary health, including the promotion of cardiovascular health and the reduction of inflammation. It was evident from the results that optimization of the Niger seed cultivation, especially from regions with favorable profiles like N/Gondar, might contribute to the improvement of nutrition and food security in Ethiopia. In doing so, any further research should be undertaken in a manner that the refinement of agricultural practices and further exploration of other varieties of the seed, while maximizing the nutritional and functional aspects of Niger seeds.

## Author Contributions

**Conceptualization:** Amare Aregahegn Dubale, Minaleshewa Atlabachew.

**Data curation:** Amare Aregahegn Dubale, Minaleshewa Atlabachew, Marie Yayinie.

**Formal analysis:** Megersa Chali Makuria, Minaleshewa Atlabachew, Marie Yayinie.

**Investigation:** Megersa Chali Makuria, Amare Aregahegn Dubale, Minaleshewa Atlabachew.

**Methodology:** Megersa Chali Makuria, Amare Aregahegn Dubale, Minaleshewa Atlabachew.

**Project administration:** Amare Aregahegn Dubale, Minaleshewa Atlabachew.

**Resources:** Amare Aregahegn Dubale, Minaleshewa Atlabachew.

**Software:** Minaleshewa Atlabachew, Marie Yayinie.

**Supervision:** Amare Aregahegn Dubale, Minaleshewa Atlabachew, Marie Yayinie.

**Validation:** Megersa Chali Makuria, Amare Aregahegn Dubale, Minaleshewa Atlabachew, Marie Yayinie.

**Visualization:** Megersa Chali Makuria, Minaleshewa Atlabachew, Marie Yayinie.

**Writing – original draft:** Megersa Chali Makuria, Marie Yayinie.

**Writing – review & editing:** Minaleshewa Atlabachew, Marie Yayinie.

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
