## [Decision Letter · Decision Letter 0]

21 Aug 2024

PONE-D-24-16831Fatty acid composition, total phenolic and total flavonoid contents, and antioxidant activity of Niger Seed (Guizotia Abyssinica) accessions collected from major producer areas of Ethiopia.PLOS ONE

Dear Dr. Yayinie,

Thank you for submitting your manuscript to PLOS ONE. After careful consideration, we feel that it has merit but does not fully meet PLOS ONE’s publication criteria as it currently stands. Therefore, we invite you to submit a revised version of the manuscript that addresses the points raised during the review process.

We look forward to receiving your revised manuscript.

Kind regards,

Awatif Abid Al-Judaibi, PhD

Academic Editor

PLOS ONE

Journal Requirements:

Reviewers' comments:

Reviewer's Responses to Questions

**Comments to the Author**

1. Is the manuscript technically sound, and do the data support the conclusions?

Reviewer #1: Yes

Reviewer #2: No

Reviewer #3: Partly

2. Has the statistical analysis been performed appropriately and rigorously? 

Reviewer #1: No

Reviewer #2: I Don't Know

Reviewer #3: Yes

3. Have the authors made all data underlying the findings in their manuscript fully available?

Reviewer #1: Yes

Reviewer #2: No

Reviewer #3: Yes

4. Is the manuscript presented in an intelligible fashion and written in standard English?

Reviewer #1: Yes

Reviewer #2: No

Reviewer #3: No

5. Review Comments to the Author

Reviewer #1: This paper addresses the extraction method, compositions, and antioxidant activity assessment of Niger Seed in different regions and zones of Ethiopia. However, there is a lack of detailed exploration of the correlation between the compositions of Niger seed and their antioxidant potential, as well as their prospective applications. Furthermore, the paper falls short of providing adequate supporting evidence for the significant claims put forth.

• The author is advised to include additional content in the paper, such as providing more background information on topics like saturated and unsaturated fatty acids, exploring Niger seed applications as inactive ingredients, and correlating their composition can be applied in various industrial sectors i.e pharmaceuticals, hygiene care, and food.

• Furthermore, it is suggested that the authors incorporate insights into the differences in localities, climates, locations, and soil types among regions and zones to better elaborate on the relationship between Niger seed composition and specific geographical factors. Expand the discussion from the results obtained, and consider reporting the calibration curve as supplementary data rather than part of the main results.

• Recommended to address the ethics statement and approval regarding field sampling permits.

• The distinction between 80% and 100% methanol extraction is noted in the scavenging antioxidant, total phenolic content, and total flavonoid content but not in the fatty acid compositions, prompting a query about potential differences in this aspect.

• Additionally, the study mentions the poor correlation of flavonoids and phenolics contents with antioxidant activity, suggesting the need to support this observation with additional scientific reports to support the findings.

• There is a lack of reporting on the statistical analysis in Tables 4 and 5, necessitating a careful interpretation of each superscript (a-d) from the Tukey test analysis to clarify the comparators for the readers. A detailed explanation of the statistical findings is essential to enhance the understanding of the results presented in the tables.

• Based on these preliminary findings, the authors are encouraged to outline future research that can stem from the analysis and discuss the potential implications.

Minor comments:

• Table 2 does not directly align with critical aspects or does not contribute significantly to the understanding of the composition of Niger Seed; it may be deemed irrelevant in the context of this paper.

• There have been identified typographical, spelling, and grammatical errors that require careful attention. For instance, on Page 8, the term "IC50%", representing the inhibitory concentration, should not be presented as a percentage since it pertains to concentration or dosage. On Page 6, the phrase "sum upping" would be more appropriately replaced with "summing up."

• Page 4; Materials and methods (Na2WO4.2H2O) is Sodium tungstate dehydrate.

Reviewer #2: The original and interesting aspect to this work relates to sampling from different geographical/climatic regions, however the authors fail to make a strong case for this. The stated "core target" of this study was the assessment of the nutritional components of niger seed (Guizotia abyssinica) based on geographical origin, cultivation season and seed variety however the authors didn't establish how they intended to do address each of these questions nor how their findings related to their stated objectives. A more logical introduction setting out your objectives, structured study design including a map of sampling locations to assist the reader would have been helpful.

What seed varieties were sampled at each sampling site? Could differences in variety or seasonal variations account for your observations rather than geographical location, without this information we cannot draw any firm conclusions from your findings.

Data from one of their five sampling sites (HG/Wolega) falls well outside the expected range for fatty acid content in Guizotia abyssinica based on previously published findings and this discrepancy has not been be accounted for.

The Ferric Ion Reducing Antioxidant Power (FRAP) is a more reliable assay for assessment of antioxidant activity and should't be too difficult to establish in your lab.

The manuscript would have benefited from professional editing prior to submission.

I have been unable to find any supporting information files despite authors stating they are included in the submission.

Reviewer #3: You have to track the mechanism of action using suitable computational techniques to support your data, such as network pharmacology, molecular dynamic simulation.

More biological assays are required

comparable studies with other species could be useful for data support

6. PLOS authors have the option to publish the peer review history of their article (what does this mean?). If published, this will include your full peer review and any attached files.

Reviewer #1: No

Reviewer #2: No

Reviewer #3: No

---

## [Author Response · Author response to Decision Letter 0]

5 Oct 2024

We have given detailed response and attached it

---

## [Decision Letter · Decision Letter 1]

20 Nov 2024

PONE-D-24-16831R1Fatty acid composition, total phenolic and total flavonoid contents, and antioxidant activity of Niger Seed (Guizotia Abyssinica) accessions collected from major producer areas of Ethiopia.PLOS ONE

Dear Dr. Yayinie,

Thank you for submitting your manuscript to PLOS ONE. After careful consideration, we feel that it has merit but does not fully meet PLOS ONE’s publication criteria as it currently stands. Therefore, we invite you to submit a revised version of the manuscript that addresses the points raised during the review process.

We look forward to receiving your revised manuscript.

Kind regards,

Awatif Abid Al-Judaibi, PhD

Academic Editor

PLOS ONE

Journal Requirements:

Reviewers' comments:

Reviewer's Responses to Questions

**Comments to the Author**

1. If the authors have adequately addressed your comments raised in a previous round of review and you feel that this manuscript is now acceptable for publication, you may indicate that here to bypass the “Comments to the Author” section, enter your conflict of interest statement in the “Confidential to Editor” section, and submit your "Accept" recommendation.

Reviewer #4: All comments have been addressed

Reviewer #5: (No Response)

2. Is the manuscript technically sound, and do the data support the conclusions?

Reviewer #4: Yes

Reviewer #5: Yes

3. Has the statistical analysis been performed appropriately and rigorously? 

Reviewer #4: Yes

Reviewer #5: Yes

4. Have the authors made all data underlying the findings in their manuscript fully available?

Reviewer #4: Yes

Reviewer #5: Yes

5. Is the manuscript presented in an intelligible fashion and written in standard English?

Reviewer #4: No

Reviewer #5: Yes

6. Review Comments to the Author

Reviewer #4: 1. The paper still contains some spelling error. Please thoroughly check every words.

2. I think it would be better if you provided separate tables for saturated and unsaturated fatty acid profiles or a single table with clear mentions of total saturated and total unsaturated fatty acid content to highlight the importance of Niger Seeds in nutritional point of view.

Reviewer #5: The paper presents a good comparative study of four main characteristics of Niger seed oil.

The manuscript already underwent thorough revision and most of the comments held against the original version have been solved already.

However, several comments and advices to the authors still can be offered:

1) The content and ratio of fatty acids, phenolics and antioxidants in seeds and seed oil is often different from those in the oil, as known ie. for Moringa oleifera https://doi.org/10.1016/j.sajb.2023.02.010 or even depend on oil production method. Was there an attempt to analyse also oil made from the samples? Were the results comparable? How about the effect of oil preparation method?

2) In the age of green solvents and sustainable sources, methanol-based solvent system is recommended for the extraction. While, no doubt, it is efficient and relatively low-cost, the method will be hard to apply for routine quality monitoring in most of modern countries due to environmental issues? Was there an attempt to use more environmentally friendly extractants, but the yield was too low? Did you try to optimize the conditions for another solvent? If so, please add the information for the benefit of others.

7. PLOS authors have the option to publish the peer review history of their article (what does this mean?). If published, this will include your full peer review and any attached files.

Reviewer #4: **Yes: **Sadia Akter

Reviewer #5: No

---

## [Author Response · Author response to Decision Letter 1]

17 Dec 2024

We have attached on a separate document by giving a file name "Response to Reviewers"

---

## [Decision Letter · Decision Letter 2]

29 Dec 2024

Fatty acid composition, total phenolic and total flavonoid contents, and antioxidant activity of Niger Seed (Guizotia Abyssinica) accessions collected from major producer areas of Ethiopia.

PONE-D-24-16831R2

Dear Dr. Marie Yayinie,

We’re pleased to inform you that your manuscript has been judged scientifically suitable for publication and will be formally accepted for publication once it meets all outstanding technical requirements.

Kind regards,

Awatif Abid Al-Judaibi, PhD

Academic Editor

PLOS ONE

Reviewers' comments:

Reviewer's Responses to Questions

**Comments to the Author**

1. If the authors have adequately addressed your comments raised in a previous round of review and you feel that this manuscript is now acceptable for publication, you may indicate that here to bypass the “Comments to the Author” section, enter your conflict of interest statement in the “Confidential to Editor” section, and submit your "Accept" recommendation.

Reviewer #4: All comments have been addressed

Reviewer #5: All comments have been addressed

2. Is the manuscript technically sound, and do the data support the conclusions?

Reviewer #4: Yes

Reviewer #5: Yes

3. Has the statistical analysis been performed appropriately and rigorously? 

Reviewer #4: Yes

Reviewer #5: N/A

4. Have the authors made all data underlying the findings in their manuscript fully available?

Reviewer #4: Yes

Reviewer #5: Yes

5. Is the manuscript presented in an intelligible fashion and written in standard English?

Reviewer #4: Yes

Reviewer #5: Yes

6. Review Comments to the Author

Reviewer #4: 1. A comparative analysis of Niger seed composition with other commonly consumed oilseeds would provide valuable context and highlight its unique nutritional attributes.

2. The study could benefit from a discussion of future research directions, such as investigating the potential health benefits of Niger seed in in vivo or clinical studies.

Amidst these suggestions, the study provides a thorough analysis of key compositional aspects of Niger seed, including fatty acid profiling, TPC, TFC, and AA assessment. This multifaceted approach offers valuable insights into the seed's potential health benefits. I think the research paper is ready to be published.

Reviewer #5: While the manuscript was properly revised, it could use a little bit of polishing touch, like changing the phrase "These result indicates" to either "This result indicates..." or "These results indicate...". Please invest a bit more time to smooth the text.

7. PLOS authors have the option to publish the peer review history of their article (what does this mean?). If published, this will include your full peer review and any attached files.

Reviewer #4: **Yes: **Sadia Akter

Reviewer #5: No

---

## [Editor Report · Acceptance letter]

2 Jan 2025

PONE-D-24-16831R2 

PLOS ONE

Dear Dr. Yayinie, 

I'm pleased to inform you that your manuscript has been deemed suitable for publication in PLOS ONE. Congratulations! Your manuscript is now being handed over to our production team.

Kind regards, 

on behalf of

Professor Awatif Abid Al-Judaibi 

Academic Editor

PLOS ONE